# Determination of the Elements Composition in *Sempervivum tectorum* L. from Bulgaria

**Galia Gentscheva** [1,2], **Irina Karadjova** [3], **Poli Radusheva** [4], **Stefka Minkova** [4], **Krastena Nikolova** [4,*], **Yoana Sotirova** [5] , **Ina Yotkovska** [1] **and Velichka Andonova** [5,*]

1  Department of Chemistry and Biochemistry, Medical University of Pleven, 5800 Pleven, Bulgaria; gentscheva@mu-pleven.bg (G.G.); ina.iotkovska@mu-pleven.bg (I.Y.)
2  Institute of General and Inorganic Chemistry, Bulgarian Academy of Sciences, 1113 Sofia, Bulgaria
3  Faculty of Chemistry and Pharmacy, University of Sofia, 1164 Sofia, Bulgaria; Karadjova@chem.uni-sofia.bg
4  Department of Physics and Biophysics, Medical University of Varna, 9000 Varna, Bulgaria; radusheva@mu-varna.bg (P.R.); stefka.minkova@yahoo.com (S.M.)
5  Department of Pharmaceutical Technologies, Medical University of Varna, 9000 Varna, Bulgaria; Yoana.Sotirova@mu-varna.bg
*  Correspondence: kr.nikolova@abv.bg (K.N.); velichka.andonova@mu-varna.bg (V.A.); Tel.: +35-988-860-3272 (K.N.); +35-989-783-2753 (V.A.)

**Abstract:** *Sempervivum tectorum* L. is an evergreen plant with fleshy blue-green leaves forming a rosette. The plant is well-known in alternative medicine and has been used for thousands of years. Traditionally for medicinal purposes, the plant is used as a juice obtained by simple squeezing of fresh plants leaves. The total content of Ca, K, Na, Mg, Mn, Fe, Zn, Cu, Co, Al, V, Cr, Ni, Mo, Ba, Pb, Cd, Hg, As, and Tl in plant leaves of *Sempervivum tectorum* L. sampled from different habitats in Bulgaria was determined after microwave digestion and measurements by inductively coupled plasma mass spectrometry and flame atomic absorption spectrometry. Furthermore, the bioavailable fraction of essential elements Ca, Mg, Fe, Mn, and Zn was defined after extraction with a hydrochloric acid solution, mimicking stomach juice. The total element content showed a high bioavailability of essential human health elements, such as Ca, Mg, Fe, and Zn. Additionally, essential and toxic elements concentrations were quantified in a fresh juice, obtained by squeezing from plant leaves, as most frequently used in folk medicine. The results obtained demonstrated high concentrations of K, Mg, Ca, Zn, and Cu, which could be accepted as an explanation and a further confirmation of the anti-inflammatory action of this plant.

**Keywords:** *Sempervivum tectorum* L.; mineral content; bioavailable fraction; heavy metals

## 1. Introduction

There are over 3000 plant species in Bulgaria, of which more than 600 are used for medicinal purposes. Bulgarian herbs contain a high percentage of biologically active substances [1]. They are rich in various chemical compounds: alkaloids, glycosides, saponins, polysaccharides, tannins, flavonoids, coumarins, essential oils, vitamins, and trace elements. The pharmacological and medicinal action and application of Bulgarian herbs has been an important topic and subject of many studies. One of the most outstanding achievements of the Bulgarian pharmaceutical industry was the creation of the medicinal preparation "Nivalin" by Prof. D. Paskov. The active substance of which is the alkaloid galantamine, extracted from *Leucojum aestivum* L. [2]. Most of the achievements of contemporary medical science are based on bioactive compounds extracted from medicinal plants [3].

*Sempervivum tectorum* L. (synonym: Sempervivum tectorum var. arvernense (Lecoq & Lamotte) Zonn., Sempervivum tectorum var. andreanum (Wale) O.Bolòs & Vigo) belongs to a large family of Crassulaceae with crassulacean acid metabolism, native to the mountains of southern Europe and cultivated in the whole of Europe.

*Sempervivum tectorum* L. (houseleek) is an evergreen plant with fleshy blue-green leaves forming a rosette, which grows on dry to fresh sandy soils and in sunny to semi-sunny places. The plant is well-known in folk medicine and has been used for thousands of years.

In recent years, researchers have focused on studies of the characterization of the bioactive ingredients of this plant and their ability to restore liver function [4], their antioxidant properties [5], the potential for wound healing [6], anti-inflammatory action, and analgesic and detoxicating properties [7–9]. Most of these properties of *Sempervivum tectorum* L. are connected with phenolic compounds identified in fresh juices squeezed from plant leaves [10,11]. However, according to the author's knowledge, despite of the numerous uses of *Sempervivum tectorum* L. in folk medicine, it remains poorly known from the viewpoint of systematic investigations into trace element content, element bioavailability, and correlation between essential element content and antioxidant activity.

The trace element content is an essential characteristic of any plant. However, there are no such data for *Sempervivum tectorum* L., neither for environmental safety, nor the effect on human health. The objective of this study was to investigate the level of the elements Ca, K, Na, Mg, Mn, Fe, Zn, Cu, Co, Al, V, Cr, Ni, Mo, Ba. Pb, Cd, Hg, and As as a total content in plant samples of *Sempervivum tectorum* L. obtained from different sampling sites; natural and affected at different levels by human activities. The urban soil (A) is from an urban park close to center of the city, and the fertilized soil (C) is from land used for agriculture purposes for many years. The rural soil (B) and mountain soil (D) might be accepted as natural; however, with varying composition. The bioavailable fraction of essential elements Ca, Mg, Fe, Mn, and Zn, defined according to a standard procedure for element migration in hydrochloric acid that mimics food digestion processes in the stomach, was quantified. Additionally, K, Ca, Mg, Na, Fe, Mn, Zn, Al, Cu, and Cr were determined in the freshly squeezed juice from plant leaves, as directly used in folk medicine against ear pain.

## 2. Materials and Methods

*Plant Material*. *S. tectorum* plants were from different habitats, grown on city soils (A), village soils (B), fertilized soils (C), and mountain soils (D). The leaves of the plants were removed, thoroughly washed with deionized water to remove all possible external contaminants, and used immediately for:

(i).     the preparation of fresh juice after squeezing;
(ii).    the preparation of a fresh homogeneous sample mix after milling for the determination of bioavailable fraction;
(iii).   the preparation of dry mass after oven drying to a constant weight at 40 °C and homogenization by careful grinding.

Sample preparation before analysis:

*Reagents*: 67% $HNO_3$ (supra pure, Merck, Darmstadt Germany); 30% $H_2O_2$ (supra pure, Merck, Darmstadt Germany); 37% HCl (p.a. Sigma-Aldrich, Darmstadt Germany).

### 2.1. Determination of Total Content of Elements

A dry sample of around 0.5 g was weighed in Teflon vessels of a microwave digestion system, 6 mL 67% $HNO_3$ and 2 mL 30% $H_2O_2$ were added, and samples were left overnight. Microwave digestion was performed for 20 min: 10 min to reach 180 °C and 10 min maintained at this temperature. After cooling, samples were transferred to a 50 mL volumetric flask and diluted up to the mark with deionized water. A blank sample was passed through the whole analytical procedure.

### 2.2. Determination of Bioavailable Fraction in Fresh Leaves

A sample of 2.0 g of fresh leaves was milled with 50 mL deionized water in a plastic container. After that, 50 mL 0.14 mol $L^{-1}$ HCl were added, and the mixture was shaken for at least 1 min. The suspension was left for several minutes to settle, and the pH of the clear supernatant was measured. If the pH was above 1.5, 2 mol $L^{-1}$ HCl solution

was added drop-wise while mixing until the pH reached values between 1.0 and 1.5. The container was closed and agitated at $37 \pm 2$ °C for 1 h. After that, the suspension was left for a further 1 h at $37 \pm 2$ °C. The mixture was protected from daylight. The solid matter was separated by centrifugation and, if necessary, filtrated through a membrane filter (0.22 μm) to remove all solid particles. The separation should be completed as soon as possible after completing the standing time; centrifuging should take no longer than 10 min. Next, the obtained solution was evaporated on a hot plate to 2–3 mL, 3 mL of conc. $HNO_3$ was added for digestion of the organic components, and, finally, the sample was quantitatively transferred to a 25 mL flask and made up with deionized water [12].

### 2.3. Determination of Elements in Juice Obtained by Squeezing of Fresh Leaves

A sample of 2.0 g juice (obtained after filtration of fresh juice through a 0.22 μm membrane filter) was transferred in a glass beaker and treated with 1 mL 67% $HNO_3$ on a hot plate. After 1 h, the solution was cooled and diluted in a 10 mL volumetric flask with distilled water.

Apparatus for quantitative measurement of chemical elements:

Flame atomic absorption spectrometry: The content of Fe, K, Mn, Mg, Na, and Zn was measured by flame atomic absorption spectrometry (Thermo Electron—SOLAAR Mkll M5 series, UK) in an air/acetylene flame under optimized instrumental parameters. The content of Ca was measured in $N_2O$/acetylene flame, using the same instrument. Stock standard solutions of Ca, Fe, K, Mn, Mg, Na, and Zn ($1.000$ g $L^{-1}$(Merck)) were used for the preparation of diluted working standards.

Inductively coupled plasma mass spectrometry: The content of As, Al, Ba, Cd, Co, Cr, Cu, Hg, Mo, Ni, Pb, and V was measured by ICP-MS using an inductively coupled plasma mass spectrometer "X SERIES 2"—Thermo Scientific, USA with a 3 channel peristaltic pump; concentric nebulizer; Peltier-cooled spray chamber (4 °C); Xt interface option; Ni cones. Optimized instrumental parameters: forward plasma power of 1400 W; plasma gas flow 13 L min/L; nebulizer flow 0.85 L/min; dwell time 30 ms; measurements $3 \times 30$ scans. Stock standard solutions: multielement standard solution 5 for ICP (TraceCERT®, Merck), 1000 mg/L As (Fluka, Sigma-Aldrich) and 1000 mg/L Hg (Fluka, Sigma-Aldrich) were used for the preparation of diluted working standard solutions for calibration of ICP-MS.

The accuracy of the analytical procedure used was validated by the analysis of certified reference material NIST SRM 1573a Tomato leaves. The very good agreement with the certified values and the recoveries above 95% achieved for all certified elements confirmed the reliability of the results obtained for total element contents (see Table 1). Limit of detection and limit of quantification was calculated for each element based on standard deviation of blanks sample for the respective procedures using 3 σ criterium (LOD) and 10 σ criterium (LOQ). Calculated values for LOD and LOQ are presented in Table 1.

### 2.4. Statistical Analysis

Data for the concentrations of chemical elements were processed to obtain the mean and standard deviation of the mean (SD). One-way analysis of variance, followed by a Student's *t*-test was used to compare the mean values. A value of $p < 0.05$ was considered to be statistically significant.

**Table 1.** Results and recoveries for chemical element contents (mg/kg) determined in NIST SRM 1573a Tomato leaves (three parallel determinations).

| Element, mg/kg | Determined (Mean $\pm$ sd) | Certified (Mean $\pm$ sd) | Recovery, % (Mean) | LOQ/LOD, mg/kg |
|---|---|---|---|---|
| Al (ICP-MS) | 594 $\pm$ 4 | 598.4 $\pm$ 7.1 | 99.3 | 0.10/0.35 |
| As (ICP-MS) | 0.1088 $\pm$ 0.056 | 0.1126 $\pm$ 0.0024 | 96.6 | 0.02/0.06 |
| Cd (ICP-MS) | 1.456 $\pm$ 0.016 | 1.517 $\pm$ 0.027 | 96 | 0.02/0.05 |
| Ca (FAAS) | 49,441 $\pm$ 342 | 50,450 $\pm$ 550 | 98 | 2/6 |
| Cr (ICP-MS) | 1.92 $\pm$ 0.04 | 1.988 $\pm$ 0.034 | 96.6 | 0.05/0.15 |
| Co (ICP-MS) | 0.5588 $\pm$ 0.021 | 0.5773 $\pm$ 0.0071 | 96.8 | 0.02/0.06 |
| Cu (ICP-MS) | 4.56 $\pm$ 0.11 | 4.70 $\pm$ 0.14 | 97 | 0.1/0.3 |
| Fe (FAAS) | 363.8 $\pm$ 2.1 | 367.5 $\pm$ 4.3 | 99 | 3/10 |
| Mn (FAAS) | 243.8 $\pm$ 9.3 | 246.3 $\pm$ 7.1 | 99 | 3/10 |
| Hg (ICP-MS) | 0.0329 $\pm$ 0.0043 | 0.0341 $\pm$ 0.0015 | 96.5 | 0.02/0.06 |
| Ni (ICP-MS) | 1.536 $\pm$ 0.031 | 1.582 $\pm$ 0.041 | 97.1 | 0.02/0.05 |
| K (FAAS) | 26,490 $\pm$ 312 | 26,760 $\pm$ 480 | 99 | 5/15 |
| Na (FAAS) | 134.3 $\pm$ 2.5 | 136.1 $\pm$ 3.7 | 98,7 | 5/15 |
| V (ICP-MS) | 0.809 $\pm$ 0.042 | 0.835 $\pm$ 0.034 | 96.9 | 0.02/0.06 |
| Zn (FAAS) | 30.02 $\pm$ 0.56 | 30.94 $\pm$ 0.55 | 97 | 1/3 |

## 3. Results

The profile of chemical elements in plants depends on the geochemical characteristics of the soil [13] and on the ability of plants to selectively accumulate minerals essential for their growth. For given plants, the content of mineral and trace elements is characteristic and will be affected by different factors, such as the physical and chemical properties of the soil, application of natural and artificial fertilizers, and climatic conditions of the region. The results obtained for the total content of elements in *Sempervivum tectorum* L. are presented in Tables 2–4.

**Table 2.** Essential (basic) element contents in dry samples.

| | K g kg$^{-1}$ | Ca g kg$^{-1}$ | Mg g kg$^{-1}$ | Na mg kg$^{-1}$ | Fe mg kg$^{-1}$ | Mn mg kg$^{-1}$ | Zn mg kg$^{-1}$ |
|---|---|---|---|---|---|---|---|
| **city soils (A), number of plant samples-4** | | | | | | | |
| mean | 18.0 | 107 | 10.6 | 93.6 | 325 | 35.4 | 49.2 |
| min | 9.36 | 102 | 4.39 | 16.4 | 188 | 13.9 | 45.1 |
| max | 29.9 | 115 | 12.3 | 234.9 | 398 | 65.4 | 51.7 |
| **village soils (B), number of plant samples-5** | | | | | | | |
| mean | 11.1 | 116 | 11.6 | 206 | 384 | 30.7 | 79.0 |
| min | 7.59 | 84.7 | 7.71 | 176 | 328 | 17.4 | 42.2 |
| max | 12.9 | 132 | 18.2 | 230 | 491 | 50.9 | 135 |
| **fertilized soils (C), number of plant samples-4** | | | | | | | |
| mean | 26.3 | 66.2 | 5.97 | 74.4 | 358 | 273 | 30.5 |
| min | 10.7 | 60.7 | 3.4 | 67.2 | 243 | 102 | 26.7 |
| max | 31.4 | 103 | 7.81 | 112 | 427 | 283 | 44.8 |
| **mountain soils (D), number of plant samples-4** | | | | | | | |
| mean | 15.7 | 61.0 | 5.68 | 125 | 247 | 12.1 | 29.2 |
| min | 7.21 | 57.3 | 3.84 | 102 | 197 | 10.5 | 25.5 |
| max | 21.3 | 85.1 | 10.5 | 131 | 343 | 18.4 | 37.8 |

**Table 3.** The content (mg/kg) of non-essential elements in dry samples.

|  | Al mg kg⁻¹ | Co mg kg⁻¹ | Cu mg kg⁻¹ | Ba mg kg⁻¹ | Mo mg kg⁻¹ | V mg kg⁻¹ | Cr mg kg⁻¹ |
|---|---|---|---|---|---|---|---|
| **A** | | | | | | | |
| mean | 23.3 | 0.46 | 7.91 | 51.0 | 0.86 | <0.02 | 0.42 |
| min | 17.0 | 0.32 | 5.63 | 48.5 | <0.02 * | <0.02 | 0.37 |
| max | 32.6 | 0.56 | 11.0 | 53.9 | 2.53 | <0.02 | 0.45 |
| **B** | | | | | | | |
| mean | 61.1 | 0.39 | 8.12 | 65.4 | 1.96 | 0.10 | 0.63 |
| min | 38.5 | 0.35 | 5.33 | 50.6 | <0.02 | <0.02 | 0.49 |
| max | 99.6 | 0.47 | 10.7 | 74.0 | 5.62 | 0.23 | 0.90 |
| **C** | | | | | | | |
| mean | 257.6 | 2.13 | 9.14 | 145.8 | <0.02 | 0.05 | 0.76 |
| min | 94.5 | 1.12 | 7.43 | 85.3 | <0.02 | <0.02 | 0.37 |
| max | 301.2 | 2.54 | 12.32 | 153.2 | <0.02 | 0.17 | 0.94 |
| **D** | | | | | | | |
| mean | 18.5 | 0.24 | 7.32 | 38.7 | <0.02 | <0.02 | 0.38 |
| min | 13.4 | 0.05 | 4.91 | 29.5 | <0.02 | <0.02 | 0.23 |
| max | 21.3 | 0.32 | 8.94 | 50.4 | <0.02 | <0.02 | 0.42 |

* Limit of detection.

**Table 4.** The total content (mg/kg) of toxic elements (Cd, Pb, As, Hg, and Ni) in dry samples.

|  | Cd mg kg⁻¹ | Pb mg kg⁻¹ | As mg kg⁻¹ | Hg mg kg⁻¹ | Ni mg kg⁻¹ |
|---|---|---|---|---|---|
| **A** | | | | | |
| mean | 0.27 | 2.66 | 0.14 | 0.05 | 2.03 |
| min | 0.17 | 1.56 | <0.02 | <0.02 | 1.32 |
| max | 0.46 | 3.99 | 0.36 | 0.10 | 2.38 |
| **B** | | | | | |
| mean | 0.23 | 1.05 | 0.05 | 0.03 | 2.40 |
| min | <0.02 | 0.63 | 0.03 | <0.02 | 2.19 |
| max | 0.26 | 1.42 | 0.09 | 0.05 | 2.73 |
| **C** | | | | | |
| mean | 0.10 | 3.18 | 0.07 | 0.03 | 4.51 |
| min | <0.02 | 1.43 | <0.02 | <0.02 | 1.29 |
| max | 0.27 | 4.02 | 0.12 | 0.05 | 4.78 |
| **D** | | | | | |
| mean | <0.02 | 1.29 | 0.08 | 0.02 | 0.89 |
| min | <0.02 | 0.54 | <0.02 | <0.02 | 0.32 |
| max | <0.02 | 1.78 | 0.11 | 0.05 | 1.15 |

The bioavailable content of Ca, Mg, Zn, Mn, and Fe in fresh leaves of *Sempervivum tectorum* L. is depicted in Table 5.

The concentrations of elements in juice obtained from fresh leaves are presented in Table 6.

**Table 5.** Bioavailable content of Ca, Mg, Zn, Mn, and Fe in fresh leaves of *Sempervivum tectorum* L. as a mean values (RSD for all samples varied between 3–8%).

|   | Ca g kg$^{-1}$ | Mg g kg$^{-1}$ | Zn mg kg$^{-1}$ | Mn mg kg$^{-1}$ | Fe mg kg$^{-1}$ |
|---|---|---|---|---|---|
| **B** | 5.07 | 0.40 | 3.55 | 1.97 | 16.5 |
| **C** | 2.95 | 0.44 | 3.58 | 9.02 | 20.0 |
| **D** | 3.16 | 0.37 | 2.37 | 1.71 | 12.3 |

**Table 6.** Element concentrations (mg/L) in fresh juice from *Sempervivum tectorum* L.

| Elements | A | Elements | A |
|---|---|---|---|
| K, mg L$^{-1}$ | 133 | Na, mg L$^{-1}$ | 0.7 |
| Ca, mg L$^{-1}$ | 561 | Zn, mg L$^{-1}$ | 1.95 |
| Mg, mg L$^{-1}$ | 2845 | Al, mg L$^{-1}$ | 3.45 |
| Fe, mg L$^{-1}$ | 0.07 | Cu, mg L$^{-1}$ | 0.28 |
| Mn, mg L$^{-1}$ | 2.40 | Cr, mg L$^{-1}$ | 0.29 |

## 4. Discussion

As seen from Table 1, *Sempervivum tectorum* L. contains an extremely high calcium content, exceeding by between 3–10 times the concentrations of the second highest content element, K. No statistically significant differences were found for the total content of essential elements (except Ca and Mn) in plants from different regions, confirming the bio-uptake ability of plant toward essential elements [14]. Unexpectedly a higher total content of Ca was observed in rural and urban plants in comparison with plants from fertilized and mountain regions. Significantly higher concentrations of Mn were determined in the plants grown on fertilized soils, which might be explained by the high bioavailable Mn content in these soils, as the same concentrations were measured in other herbs from the same region. The total content of essential elements presented at lower concentration levels in *Sempervivum tectorum* L. is close to the content of these elements in other herbs from these regions [15,16].

Table 2 lists the results obtained for some nonessential elements. As can be seen, the plants grown on agricultural (fertilized) soils differed from the others with their higher concentrations of Al, Co, and Ba. However, only the concentrations of cobalt were surprising, as Al of such and higher concentrations is found in herbs from this region [15].

Another critical aspect is the good quality control of medicinal herbs, to protect consumers from contamination, as many medicinal herbs and their mixtures can present a health risk due to toxic elements [17].

Toxic element levels in raw plant material or prepared products/extracts/infusions is regulated by documents at global, national, or regional level. Strict control of contaminant levels and their minimization is required by the World Health Organization (WHO) through guidelines such as the good agricultural and collection practices (GACP) for medicinal plants and good manufacturing practices (GMP) for herbal medicines. Maximal values for toxic elements in herbal drugs and extracts have been discussed and compared by several authors [18–20].

According to the World Health Organization, cadmium concentrations and lead in herbal medicines and products are regulated at 0.3 mg kg$^{-1}$ Cd and 10.0 mg kg$^{-1}$ Pb [18]. In different countries, the law sets lower limits, and a very good comparison of the various permissible limits is presented by Luo et al. [20]. As shown in Table 3, the concentrations of toxic elements meet the requirements of the WHO, and only in one single case was the cadmium concentration exceeded, for urban soil. Expectedly, the results for elements such as As, Cd, and Pb are highest in plants grown on urban soils. Most probably, in this case both soil pollution and aerosol deposition are responsible for the high toxic element content. Although, it is clearly important to harvest medicinal plants from clean sites

without anthropogenic influences such as mountain regions. A relatively high content was determined for Ni and Pb in plants grown on fertilized soils, most likely connected with Ni and Pb contamination by the phosphate fertilizers applied.

In this study, sampling for all studied plants and sampling sites was performed in the summer season, with some efforts to use plants in the same vegetation period. Taking into account that *Sempervivum tectorum* L. is a perennial plant, additional research is required to elucidate any correlation between plant age and chemical element content.

In Table 4, the results found for the operationally defined bioavailable content (see Section 2.3) of Ca, Mg, Zn, Mn, and Fe in fresh leaves of *Sempervivum tectorum* L. (after two hours of treatment in pH 1.0–1.5) are depicted. Plants growing on mountains, villages, and fertilized soils were used. The percentage content of bioavailable fraction varied between 4–14% for all studied essential elements (Table 7). It should be pointed out that the content of Ca and Fe, which might be accepted as being most responsible for the health functions of *Sempervivum tectorum* L., is almost constant in the bioavailable fractions from all samples. High concentrations of Ca in this fraction justify the use of *Sempervivum tectorum* L. as a national remedy for the treatment of gastric ulcers, possibly because of the beneficial calcification effect.

**Table 7.** Bioavailable concentrations of Ca, Mg, Zn, Mn, and Fe as a percentage of total content.

|  | Ca Bioavaible Fraction, % | Mg Bioavaible Fraction, % | Zn Bioavaible Fraction, % | Mn Bioavaible Fraction, % | Fe Bioavaible Fraction, % |
|---|---|---|---|---|---|
| **B** | 4.37 | 3.45 | 4.49 | 6.42 | 4.30 |
| **C** | 4.46 | 5.63 | 11.7 | 3.30 | 5.59 |
| **D** | 5.18 | 6.51 | 8.12 | 14.1 | 4.98 |

Although the concentration of Mn is still highest as a bioavailable concentration, the degree of extraction was significantly lower, most probably depending on the different Mn species present in the leaves. Therefore, it might be assumed that the Mn bio-uptake would be highest from agricultural (fertilized) soils, most probably because of suitable pH values.

Determination of elements in juice from fresh leaves. Fresh juice obtained by squeezing leaves from *Sempervivum tectorum* L. was widely used as folk medicine against ear pain. As shown in Table 6, this effect can most probably be explained by the high Mg concentrations, analogous to the pharmaceuticals used for external application (Mg-gels or Mg-oils) with anti-inflammatory and regenerative actions and improved blood circulation [21–23]

## 5. Conclusions

The total element contents and the bioavailable fraction of essential elements Ca, Mg, Zn, Mn, and Fe were determined in leaves of *Sempervivum tectorum* L. The control of the quality of medicinal plants used in traditional medicine and pharmacy is an important step for consumer protection from contamination and health risks. The determination of toxic element content in plants grown on different soils clearly shows the contamination of plants from urban soils and plants fertilized with phosphate fertilizers. The high bioavailable concentrations of essential elements could explain the wide use of this plant in folk medicine. For example, the high Mg content in fresh juice is responsible for its anti-inflammatory action and application as an ear pain reliever.

**Author Contributions:** K.N. constructed and conceived the project. G.G. and I.K. designed the study. P.R., S.M., Y.S. and I.Y. performed the study. V.A. and Y.S. analyzed the data. G.G. and I.K. wrote the newspaper. All authors have read and agreed to the published version of the manuscript.

**Funding:** This research was funded by the Bulgarian Ministry of Education and Science under the National Research Programme "Healthy Foods for a Strong Bio-Economy and Quality of Life" approved by DCM # 577/17.08.2018.

**Institutional Review Board Statement:** Not applicable.

**Informed Consent Statement:** Not applicable.

**Data Availability Statement:** Datasets from the time of this study are available from the respective author upon reasonable request.

**Acknowledgments:** Special thanks to Medical University–Varna for the provided financial support for paper publication.

**Conflicts of Interest:** The authors declare no conflict of interest.

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
