# Peer review of "Determination of the Elements Composition in Sempervivum tectorum L. from Bulgaria"

_horticulturae, doi:10.3390/horticulturae7090306_

Round 1

Reviewer 1 Report

The study presents the elemental composition analysis of samples obtained from different conditions of exposure to disturbance and contamination. It is an important study that contributes to the knowledge of this resource as a functional food alternative and to associate it with its therapeutic properties, however, the methodology used must be complemented with a more complete quantitative analysis.

The introduction presents a large amount of information, but does not support the aforementioned background with up-to-date references on the subject of study.

Digestion for the simultaneous analysis of transition metals and volatile metals such as Hg and As, require an additional attenuation process or chemical speciation with the formation of hydrides, which is not described in the methodology.

The quantitative analysis of the elements by ICP-MS does not indicate the construction of calibration curves using authentic standards for all the elements reported, which indicates that the analysis method was performed by full scan in ICP-MS, which is only a qualitative approximation of the elementary content, without the rigor of a linear analysis with sufficient accuracy. The clear example of this inconsistency is seen in the Pb content in the city and mountain samples.

It would be advisable to make a broader description of the methodology in the ICP-MS analysis, especially in the conditions of the equipment and the standards used.

Additionally, the following corrections are recommended:

Page 2, Line 59, The possible relationship of mineral content with antioxidant activity is mentioned, but experiments on this activity are not included, nor are more data on this property provided.

Page 2, Line 72-74. The number N of samples is not indicated for the statistical analysis described or the averages shown in tables 1, 2, 3, 4, 5 and 6.

Page 2, Line 87. The batch number of the certified material used must be indicated.

Page 5, Lines 140-142. No significant differences are observed in the groups due to the great variability of the results, which may be associated with the total scanning method used in the ICP-MS technique.

Page 5, Lines 157-164, The results shown in table 3, generate doubts due to the way of total scanning as the analysis was done.

Page 6, Lines 194-195 This conclusion is not based on the results of this work.

Page 6, Lines 195-196. This conclusion is in doubt due to the qualitative technique of the elemental content of the samples.

Reviewer 2 Report

horticulturae-1358692-peer-review-v1

Part I: Determination of the Elements Composition in Sempervivum Tectorum L. from Bulgaria

Some suggestions are given below

  • Line 46 , page 2

Please check that all the botanical names and synonyms of the reported specie are according to http://www.worldfloraonline.org/

 .

  • Please see and include the reference

http://www.worldfloraonline.org/search?query=Sempervivum+tectorum+L.&limit=24&start=0&sort=

  • In the different methodologies please mention the brand of the reagents and the company of origin, for example HNO3 (Merck, from Germany or Sigma-Aldrich or others).

  • ICP-MS under optimized instrumental parameters (In the different methodologies please mention the trademark and origin of the equipment, for example Perkin Elmer, Agilent USA or others).

  • Lines 186-189, page 6.

Please include relevant literature to support your observation.

  • Section Conclusion: Please review the conclusions, which should highlight the results found in the investigation. Be more specific.
  • Title of manuscript

The word “Part 1” is necessary?

The following paragraph does not understand the meaning of why it is there

“For example, high Mg content in fresh juice is responsible for its anti-inflammatory action and application as an ear pain reliever”

After considering suggestions, the manuscript should be accepted for publication.

Reviewer 3 Report

Please see attached files for my detailed commnets

Round 2

Reviewer 1 Report

The authors promptly addressed the recommendations on the adequate description of the techniques used. Authentic standards and independent calibration for each element indicate analytical quality in their results. In addition, the discussion and conclusions have been adjusted.

Reviewer 3 Report

Authors addresed the commments as much as they could....